# Molecular Karyotyping on *Populus simonii* × *P. nigra* and the Derived Doubled Haploid

**DOI:** 10.3390/ijms222111424

**Published:** 2021-10-22

**Authors:** Bo Liu, Sui Wang, Xiaoyan Tao, Caixia Liu, Guanzheng Qu, Quanwen Dou

**Affiliations:** 1Key Laboratory of Adaptation and Evolution of Plateau Biota, Northwest Institute of Plateau Biology, Chinese Academy of Sciences, Xining 810008, China; liubo176@mails.ucas.ac.cn (B.L.); taoxiaoyan19@mails.ucas.ac.cn (X.T.); 2State Key Laboratory of Tree Genetics and Breeding, Northeast Forestry University, Harbin 150040, China; wangsui.ws@163.com (S.W.); liucaixia2020@outlook.com (C.L.); 3Key Laboratory of Soybean Biology of Ministry of Education China, Northeast Agricultural University, Harbin 150030, China; 4College of Life Sciences, University of Chinese Academy of Sciences, Beijing 100049, China; 5Key Laboratory of Crop Molecular Breeding Qinghai Province, Northwest Institute of Plateau Biology, Chinese Academy of Sciences, Xining 810008, China

**Keywords:** *Populus*, *Populus simonii* × *P. nigra*, doubled haploid, repetitive sequence, karyotype, FISH

## Abstract

The molecular karyotype could represent the basic genetic make-up in a cell nucleus of an organism or species. A doubled haploid (DH) is a genotype formed from the chromosome doubling of haploid cells. In the present study, molecular karyotype analysis of the poplar hybrid *Populus simonii × P. nigra* (*P. xiaohei*) and the derived doubled haploids was carried out with labeled telomeres, rDNA, and two newly repetitive sequences as probes by fluorescence in situ hybridization (FISH). The tandem repeats, pPC349_XHY and pPD284_XHY, with high-sequence homology were used, and the results showed that they presented the colocalized distribution signal in chromosomes. For *P. xiaohei*, pPD284_XHY produced hybridizations in chromosomes 1, 5, 8, and 9 in the hybrid. The combination of pPD284_XHY, 45S rDNA, and 5S rDNA distinctly distinguished six pairs of chromosomes, and the three pairs of chromosomes showed a significant difference in the hybridization between homologous chromosomes. The repeat probes used produced similar FISH hybridizations in the DH; nevertheless, pPD284_XHY generated an additional hybridization site in the telomere region of chromosome 14. Moreover, two pairs of chromosomes showed differential hybridization distributions between homologous chromosomes. Comparisons of the distinguished chromosomes between hybrid and DH poplar showed that three pairs of chromosomes in the DH presented hybridization patterns that varied from those of the hybrid. The No. 8 chromosome in DH and one of the homologous chromosomes in *P. xiaohei* shared highly similar FISH patterns, which suggested the possibility of intact or mostly partial transfer of the chromosome between the hybrid and DH. Our study will contribute to understanding the genetic mechanism of chromosomal variation in *P. xiaohei* and derived DH plants.

## 1. Introduction

Poplar (*Populus* spp.), a genus in the family Salicaceae, is widely distributed in most of the Northern Hemisphere due to its rapid growth, stress resistance, and economic importance [1]. This genus has traditionally been divided into six sections based on leaf, flower, and other characteristics, but five major subclades have been recovered based on recent genetic studies [2,3]. *Populus simonii* × *P. nigra* (abbreviated in the text as *P. xiaohei*) is an artificial hybrid crossed with *P. simonii Carr* (*Tacamahaca*) as the maternal plant and *P. nigra* L. (*Aigeiros*) as the paternal plant [4]. The hybrid *P. xiaohei* presents the advantages of resistance to cold, drought, and salt and barren land adaptation, and it is widely distributed to the north of the Yellow River in China [5,6,7]. Although hybrid poplar has obvious adaptability and growth advantages, its high heterozygosity makes breeding and basic research difficult. Haploid and doubled haploid (DH) plants play an important role in breeding, genetics, molecular biology, and genomics. Additionally, these plants are adapted for mutagenesis and genetic transformation experiments, presenting the advantage of the immediate production of homozygous lines. However, it is very difficult to produce DH trees, and their long-term survival is also a great challenge. DH plants were successfully developed from the anther of *P. xiaohei* in our lab and are regarded as important germplasms for basic research and breeding.

The karyotype of a plant species, usually characterized by chromosome number and appearance, is the basic genetic information of the species. Karyotyping of a species is valuable for phylogenetic and genome composition analysis, germplasm identification, and breeding applications, among others [8,9]. *Populus* species are primarily diploids with a haploid chromosome number of 19 [10,11]. Most chromosomes of *Populus* species are small and morphologically similar, making them difficult to distinguish from one another by conventional cytological methods [12,13,14]. Fluorescence in situ hybridization (FISH) provides a powerful tool for effective chromosome identification [8]. Four pairs of chromosomes were exclusively identified using 45S rDNA, 5S rDNA, and telomere sequences as FISH probes in seven *Populus* species [15]. Moreover, the conserved physical sites of both 45S rDNA and 5S rDNA were revealed across the species [15]. *Populus* karyotypes, based on individually identified chromosomes, have been proposed by developing a complete set of 19 chromosome painting probes, based on the reference genome of the model woody plant *P. trichocarpa* [16]. They also demonstrated a remarkably conserved karyotype and no inter-chromosomal structural rearrangements on any of the 19 chromosomes among the five species. The above probes were whole-chromosome painted, and they showed the collinearity between individual chromosomes among different species. However, intrachromosomal structural variations, such as deletions, duplications, and inversions, between species were not excluded. Tandem repetitive sequences are very useful to generate FISH probes in molecular karyotyping [17]. The combination of tandem repeat-based probes can enable partial or complete chromosome identification in an organism. Chromosome polymorphisms detected by tandem repeats can be used to infer chromosomal structural variation between species or even to identify different germplasms [8]. However, information for FISH-based tandem repeats in *Populus* is still limited.

The *P. xiaohei* hybrid inherits half of the set of chromosomes from *P. simonii* and *P. nigra*. The karyotype of the hybrid should be present in heterozygosis, considering the differentiation of homologous chromosomes between the two parent species. In contrast, the karyotype of DH should be homozygous, with identical homologous chromosomes occurring by the doubling of whole chromosomes. However, how the homologous chromosomes differ, recombine, and are transferred in the *P. xiaohei* hybrid and whether chromosomal variations are generated in the DH is unknown. In this study, the cytogenetic features of the *P. xiaohei* hybrid and DH were unveiled by FISH analysis using routine and newly explored probes. The results will be useful for basic research and breeding applications of *Populus* spp.

## 2. Results

### 2.1. Homozygosity Assessment by k-mer Analysis

In total, we obtained approximately 57.2 and 54.3 Gb raw reads for the heterozygous parent and the DH plants of *Populus*, respectively. After trimming low-quality and short reads, 41.7 and 38.6 Gb clean reads were obtained. As most *Populus* species genome sizes are less than 500 Mb, we set *k* = 17 for the *k*-mer analysis. According to the *17*-mer curve (Figure 1), the curve of the heterozygous parent (*P. xiaohei*) has two distinct peaks, and the DH only has a main peak. Figure 1a shows that the height of the heterozygous peak is significantly higher than that of the main peak, and we believe that the parent is highly heterozygous. Figure 1b shows that the heterozygosity of the DH is particularly low, and we confirmed the homozygosity of the sample. Another detail in Figure 1 shows a distinct trail behind the main peak at an approximate 100× depth in both curves, resulting from repeat sequences.

### 2.2. Development of Repetitive Sequence DNA Probes in P. xiaohei

During the assembly of the draft genome of *P. xiaohei*, sequences of less than 500 bp were filtered out. Finally, a 566.1 MB preassembled version of the *P**. xiaohei* genome was obtained, with a total of 318,203 scaffolds, and the scaffold N50 was 2685 bp.

BLASTN was performed with tandem repeats, as reported by Rajagopal [18]. Two candidate tandem repeats with lengths of 129 and 109 bp were identified and named pPC349_XHY and pPD284_XHY (Table 1). The blast results showed that pPC349_XHY had 88.07 and 88.12% identity with AC210658.1 and AC213418.1 in *P. trichocarpa*, respectively. pPD284_XHY shared 85.98 and 88.12% identity with AC210658.1 and AC213418.1 in *P. trichocarpa*, respectively.

Probe pPD284_XHY was used to perform the first FISH experiment on the metaphase chromosomes of *P. xiaohei*. A total of six strong signals and two weak signals on four pairs of chromosomes were revealed, of which two strong signals were located in the centromere region of two large chromosomes, and the remaining signals were located in the telomeres or subtelomeric regions in three pairs of chromosomes. The signal on one of the chromosomes was very strong (Figure 2a). The probe pPC349_XHY was used for the second round of FISH on the same mitotic spread, and the signals of pPC349_XHY and pPD284_XHY presented a colocalized distribution across the chromosomes (Figure 2b).

### 2.3. The Molecular Karyotype of P. xiaohei

Thirty-eight chromosomes were detected in root tip cell metaphase in *P. xiaohei* by DAPI staining. Two large chromosomes and two pairs of chromosomes with satellite DNAs were distinguished, and the others were small and resembled each other (Figure 3a). Furthermore, sequential FISH was carried out for chromosome identification by using labeled repetitive sequences as probes. In the first round, the newly identified repetitive sequence pPD284_XHY and telomere sequences were used (Figure 3b), while 45S rDNA and 5S rDNA were used in the second round (Figure 3c). Thus, the molecular karyotype of *P. xiaohei* was generated by the result of the FISH test (Figure 4a). The chromosome alignment in the current karyotype followed the karyotypes of the *Populus* species proposed by Xin et al. [16], which were developed by whole-chromosome oligo-painting FISH. Chromosomes 1, 8, 14, and 17 in this study corresponded completely to the alignment number of the proposed karyotypes by distinct chromosome morphology and land markers of rDNAs. The arrangements of the others mainly refer to the arm ratio and relative length of the proposed karyotypes.

The karyotyping results showed that pPD284_XHY produced hybridization sites on four pairs of chromosomes: 1, 5, 8, and 9. Chromosome 1 included one major site around the centromere; chromosome 5 harbored one major site in subtelomeric regions of the long arms; chromosome 8 had one major site in telomeric regions of the short arm, and chromosome 8 had one major site in telomeric regions of the short arm and one major site around the centromere; chromosome 9 contained one minor site in subtelomeric regions of the long arms (Figure 4a). The 45S rDNA was physically mapped on chromosomes 8 and 14, corresponding to the satellite DNAs, whereas the hybridization intensity on chromosome 8 was stronger than on chromosome 14. The 5S rDNA was physically mapped in the subtelomeric regions on the short arms of chromosomes 17. Colocalization of pPD284_XHY and 45S rDNA partial regions on chromosome 8 was revealed simultaneously (Figure 4a). Thus, combining the FISH patterns of the above probes enabled six pairs of chromosomes, chromosomes 1, 5, 8, 9, 14, and 17, to be exclusively identified in *P. xiaohei.*

With regard to the heterozygosity of *P. xiaohei*, consistency between homologous chromosomes was compared. The results demonstrated that three of six pairs of chromosomes identified by FISH patterns had apparent polymorphisms between homologous chromosomes. Chromosome 1 showed different hybridization intensities of pPD284_XHY; chromosome 8 showed distinct differences in the hybridization intensities and distribution patterns of pPD284_XHY and the hybridization intensity of 45S rDNA between two homologous chromosomes; chromosome 14 showed varied hybridization intensities of 45S rDNA between each other (Figure 4a).

### 2.4. The Molecular Karyotype of DH

A total of 38 chromosomes were detected in the DH as in the *P. xiaohei.* However, one pair of chromosomes and one chromosome were identified as satellite chromosomes in the DH instead of two pairs in the *P. xiaohei* (Figure 3, yellow arrows). pPD284_XHY produced not only similar FISH patterns on chromosomes 1, 5, 8, and 9 in the DH as in the *P. xiaohei* but also additional new hybridization sites on the telomeric regions of the short arms of chromosome 14 (Figure 3e). The 45S rDNA on chromosomes 8 and 14 and 5S rDNA on chromosome 17 were stably detected, as in *P. xiaohei* (Figure 3f). Similarly, six pairs of chromosomes were clearly identified using probe combinations in the DH.

Karyotype homozygosity of the DH was examined further. Two pairs of chromosomes had apparent polymorphisms between homologous chromosomes. Chromosome 8 showed different hybridization intensities of pPD284_XHY in the subtelomeric regions. One chromosome 14 had a stronger hybridization intensity of both pPD284_XHY and 45S rDNA than another (Figure 4b).

### 2.5. Karyotype Comparison between P. xiaohei and DH

FISH pattern comparison between *P. xiaohei* and DH was conducted on the six pairs of chromosomes that were consistently identified in each. Three of six chromosomes, chromosomes 5, 9, and 17, displayed no apparent variation in FISH patterns (Figure 4). Chromosome 1 of DH showed a fainter pPD284_XHY hybridization intensity compared with that of *P. xiaohei.* Chromosome 14 of DH had additional pPD284_XHY hybridization sites compared with those of *P. xiaohei*, while one chromosome 14 of DH showed decreased hybridization intensities of 45S rDNA. Chromosome 8 of DH was rather similar to chromosome 8 of *P. xiaohei*, with distinct strong hybridizations of pPD284_XHY around the centromere (Figure 5).

## 3. Discussion

*P. xiaohei* inherited half of the chromosomes from the parents *P. simonii* and *P. nigra*, respectively. The expected karyotype heterozygosity should be observed in the hybrid due to the chromosome differentiation between *P. simonii* and *P. nigra*, and the degree of karyotype heterozygosity should be closely related to the degree of chromosome differentiation. Highly conserved karyotypes and chromosomal synteny in *Populus* species, revealed by chromosome painting, are remarkably maintained [16]. However, FISH analysis using 45S rDNAs revealed the variation in hybridization site number and hybridization intensities between different species [15,16,19]. In this study, three pairs of chromosomes were identified as apparently different FISH patterns, which included the variation in 45S rDNA hybridization intensities and the variations in pPD284_XHY hybridization intensities and distribution patterns in *P. xiaohei*. In particular, chromosome 8 had distinctly different distribution patterns of pPD284_XHY hybridization between two homologous chromosomes. Lacking the parental karyotypes, the species origin of the different homologous chromosomes could not be determined. However, the distinguished difference in a few homologous chromosomes suggested species-specific chromosomes.

DH inherited the haplotypes of the recombined chromosomes between *P. simonii* and *P. nigra*. The expected karyotype of the derived haploid should be homozygous, as described by *k*-mer analysis. However, a clearly identified difference in FISH patterns was revealed between homologous chromosomes in three pairs of chromosomes. The DH was developed from anther culture. A high rate of somaclonal variation was found during tissue culture, especially with chromosomal structural variation [20,21,22,23,24]. The integrity of the chromosome may be affected by cell cycle fluctuations due to the replication lag of heterochromatin during clonal culture, and chromosome rearrangements can be caused by unbalanced nucleotides [25,26]. In this study, the difference between homologous chromosomes was mainly manifested as the hybridization intensities variation of the repetitive sequences, which are usually distributed on heterochromatin regions in the plant genome [27]. This phenomenon suggests that the variations between homologous chromosomes may result from rapid amplification, deletion of repetitive sequences, or micro-chromosomal rearrangements. Moreover, it prompts that the DH not only provides purer genetic lines but also novel clonal variation in germplasm creation and breeding in *Populus*. In this study, only one DH line was analyzed due to the availability of the other DH lines currently limited by the difficulty of the plant regeneration. Further, chromosomal variation possibly induced by clonal variation can be proven in different will-developed DH lines.

The DH used in this study directly descended from *P. simonii* and *P. nigra*. The expected homologous chromosomes in the DH should be the recombination types derived from *P. simonii* and *P. nigra*. However, a few of the identified chromosomes revealed exceptional FISH patterns, which distinctly differed from the putative recombination types of the parents. One pair of chromosomes 1 and 8 carried weaker pPD284_XHY hybridizations in DH than in the parent chromosomes. Repetitive sequence deletions caused by uneven crossover during the meiosis of the parent chromosomes or by somaclonal variation can be suspected [25,26]. The heterogeneity among genome organization at all length scales or even cells may also be an important cause [28].

Chromosome 14 in the DH had additional pPD284_XHY hybridizations that were absent in the corresponding parent chromosomes. The new site may have been generated from the pre-existing pPD284_XHY by rapid amplification during tissue culture or may be a segmental translocation from other chromosomes, such as chromosome 8, which included a large portion of pPD284_XHY in telomeric regions. The FISH patterns of chromosome 8 in the DH highly resembled those one of chromosome 8 in the parent. Thus, we propose the occurrence of complete or most part transfer of the chromosome and a possible crossover decrease in chromosome 8 in the parent due to high genetic differentiation.

Highly repetitive sequences are ideal chromosomal markers for chromosome identification. Two repetitive sequences were obtained by homology searching against the *P. xiaohei* genome sequences using a 145 bp tandem repeat family identified in *Populus* species [18]. Two repetitive motifs, 109 and 129 bp, were identified in the *P. xiaohei* genome. The two motifs showed approximately 97% identity in the covered sequences. One motif was distinct from the other with an additional 20 bp. The repetitive sequences were explored in the hybrid. Whether they are repeat family members or species-specific needs to be further confirmed in future analyses. Although a conserved karyotype was revealed across *Populus* species by using single-copy oligo painting [16], repetitive sequences may represent more varied intraspecies and interspecies in *Populus* based on previous studies [15,16,19] and our results. Thus, chromosomal variation or polymorphism in or between species can be revealed by exploiting additional repeat markers. Early studies showed that the 145 bp tandem repeat accounts for 1.5% of the genome in *P. deltoids* [18]. However, repetitive elements cover over 40% of the assembled genome of *P. trichocarpa* [29]. Some powerful tools, such as RepeatExplore [30,31] and RepeatModeler2 [32] are utilized to explore tandem repeats based on next-generation sequencing data. In the future, more specific probes will be exploited, based on the reference genomes that we are assembling to identify chromosomal variation and verify homologous chromosome recombination events during meiosis of *P. xiaohei.*

## 4. Materials and Methods

### 4.1. Plant Materials

The branches of *P. xiaohei* were collected from the campus of Northeast Forestry University, and DH plants were obtained by anther culture using the method described by Liu et al. [33]. The process of tissue culture is summarized as follows. Monokaryon border-stage anther tissues were collected as explants and were transferred to induction medium. After culturing in the dark for about 30–40 days, the callus was formed. Then, the haploid callus was placed in MS + 0.2 mg/L NAA + 1 mg/L 6-Benzylaminopurine (6-BA) + 0.05 mg/L TDZ (thidiazuron) medium and cultured for 30 days, and the callus developed into hardened green mass. After two months, shoots were formed from the callus. Finally, whole plants were regenerated and transferred to the greenhouse. Plantlets were grown in pots in a greenhouse under standard conditions. At the beginning of our experiment, more than 30 haploid calli lines were obtained, but the DH line used in our experiment was the first line to regenerate a complete whole plant and grew relatively normally.

### 4.2. Chromosome Preparation

The plant material was transferred to a pot and grown for 1 month before taking out the new root tips. Root tips were pretreated in ice–water mixture at 0–4 °C in a refrigerator (4 °C) for 24 h before being fixed in 3:1 (*v*/*v*) ethanol:glacial acetic acid. Chromosome spreads and slide preparation were performed as described in Dou et al. [34].

### 4.3. k-mer Analysis and Repeat Sequence Recognition

The genomic DNA of *P. xiaohei* and DH was extracted from tender leaves using a genomic DNA kit (Biotake Corporation, Beijing, China). After qualified quality control, the DNA was sent to Oebiotech Corporation (Shanghai, China) on dry ice. Two Illumina libraries (insert sizes = 300 bp) were prepared and sequenced on the HiSeq X platform (Illumina, Sanger Diego, CA, USA) with 150 bp paired-end reads. To obtain high-quality and vector/adaptor-free reads, raw reads were checked using fastQC (v.0.11.8) and filtered using fastp (v.0.20.1). Genome Characteristics Estimation (GCE, v.1.0.2) was utilized to draw the *k*-mer distribution curve and evaluate the homozygosity of each sample. Simultaneously, the preprocessed clean reads were assembled into a draft genome using SOAPdenovo2 (v 2.04) with an adjusted *k*-mer value (*k*  =  91). According to the tandem repeat sequences of *P. deltoides* and *P. ciliata* reported by Rajagopal [18], blastn (v. 2.10.1) was used to search for homologous repeat sequences as candidates in the assembled draft genome sequence of *P. xiaohei*. To verify the identities between the identified tandem repeats and those similar sequences in other *Populus* species, the sequences were searched against the NCBI nt database utilizing blastn.

### 4.4. Probe Labeling

The repeated sequences pPD284_XHY and pPC349_XHY were synthesized by GENEWIZ (https://www.genewiz.com.cn, accessed on 6 September 2019), linked to the pUCm-T vector and transferred to *Escherichia coli* for propagation. pPD284_XHY and pPC349_XHY were amplified by polymerase chain reaction (PCR) using primer M13 (F: GTTGTAAAACGACGGCCAG; R: CAGGAAACAGCTATGAC). Two repetitive sequences, pPD284_XHY and pPC349_XHY, were labeled with Tetramethyl-rhodamine-5-dUTP (red) and Fluorescein-12-dUTP (green), respectively, as described by Dou et al. [35].

The 45S rDNA, 5S rDNA, and telomere sequences are represented by their characteristic end-labeled oligonucleotide fragments [36,37]. The 45S rDNA and telomere sequences were end-labeled with FAM (green), and 5S rDNA was end-labeled with TAMRA (red). The two probes were synthesized by Sangon Biotech (Shanghai) Co., Ltd. (Shanghai, China).

### 4.5. Fluorescence In Situ Hybridization

FISH experiments were carried out with minor modifications, as described by Dou et al. [35]. The chromosome preparations were denatured in 0.2 M NaOH and 70% ethanol for 10 min at room temperature; subsequently, they were rinsed in cold 70% ethanol (−20 °C) for 1 h and quickly air-dried. The hybridization mixture per slide (total volume = 10 μL) contained 100 ng labeled probe DNA, 50% *v/v* formamide, 2 × SSC, 10% *w/v* dextran sulfate, and 0.1 μg salmon sperm DNA. The hybridization mixture with probes was denatured in the boiled water for 5 min and immersed in ice water for at least 5 min. The hybridization with oligonucleotide fragments probes was conducted directly without denaturation. The hybridization was carried out overnight at 37 °C. A sequential FISH technique with two rounds of hybridization on the same mitotic spread was adopted in this study. The slide was washed in 2 × SSC at 42 °C three times (5 min each) in the first round of FISH, and at least 10 cells with clear signals were captured. Subsequently, the slides were washed by 1 × PBS at 42 °C to remove the coverslip and then washed twice in 2 × SSC at 42 °C (15 min each). Finally, the slides were dehydrated in cold 70% ethanol, denatured again in 70% formamide at 85 °C for 2 min, dehydrated in cold 70% ethanol, and reprobed with different probes. Chromosomes were stained with 4′,6-diamidino-2-phenylindole (DAPI) in VectaShield antifade solution. Photomicrographs were obtained with an Olympus BX-63 fluorescence microscope equipped with a DP-80 CCD camera.

## 5. Conclusions

The tandem repeats pPC349_XHY and pPD284_XHY have high homology and present a colocalized distribution on chromosomes. The combination of probes distinguished the six pairs of chromosomes in *P. xiaohei* and DH. Three pairs of chromosomes showed apparent hybridization differences between homologous chromosomes in *P. xiaohei*. Two pairs of chromosomes had differential hybridization distributions between homologous chromosomes in the DH. The repeats used produced FISH hybridizations in DH similar to those in *P. xiaohei*; nevertheless, pPD284_XHY generated an additional hybridization site in the telomere region of chromosome 14. Comparisons of the distinguished chromosomes between *P. xiaohei* and DH showed that three pairs of chromosomes in DH presented hybridization patterns that varied from those of *P. xiaohei*. Chromosome 8 in the DH and one of the homologous chromosomes in *P. xiaohei* shared highly similar FISH patterns. This result raises the possibility of the intact heredity of the chromosome.

## Figures and Tables

**Figure 1 ijms-22-11424-f001:**
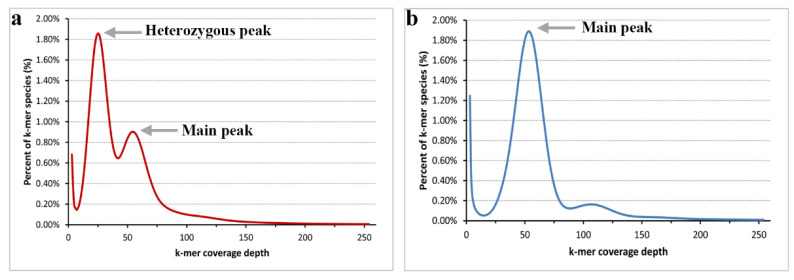
Distribution of the *17*-mer frequency. (**a**) the *k*-mer frequency distribution of the heterozygous parent; (**b**) the *k*-mer frequency distribution of the doubled haploid.

**Figure 2 ijms-22-11424-f002:**
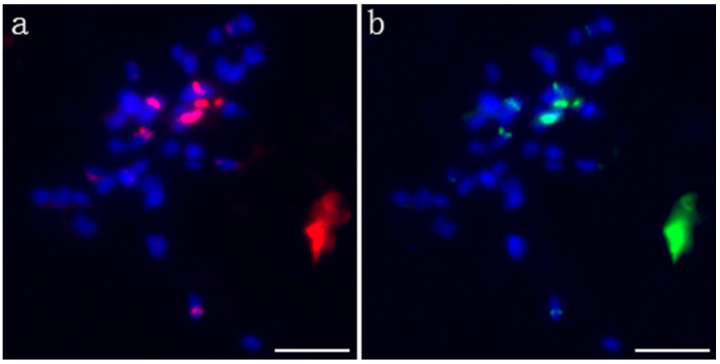
FISH patterns of pPD284_XHY and pPC349_XHY of *P. xiaohei*. (**a**) pPD284_XHY (red); (**b**) pPC349_XHY (green). Bars = 5 μm.

**Figure 3 ijms-22-11424-f003:**
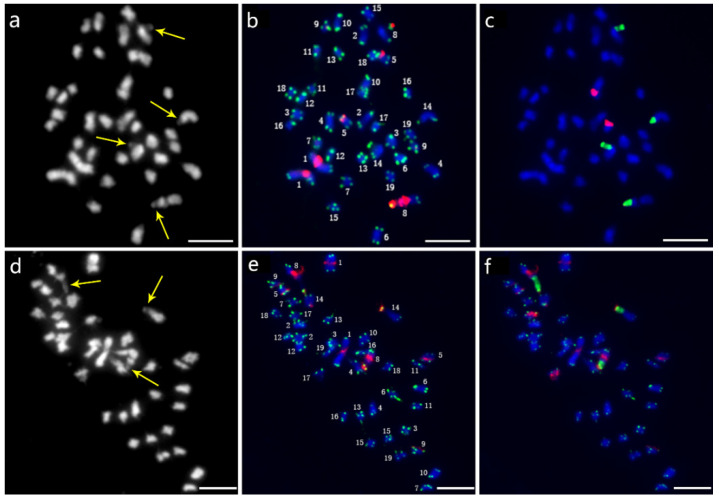
Sequential FISH patterns of *P. xiaohei* (upper panel) and the derived doubled haploid (lower panel). (**a**,**d**) Chromosome complements by DAPI staining; (**b**,**e**) FISH patterns probed with telomere sequences (green) and pPD284_XHY (red); (**c**,**f**) FISH patterns probed with 45S rDNA (green) and 5S rDNA (red). Bars = 5 μm.

**Figure 4 ijms-22-11424-f004:**
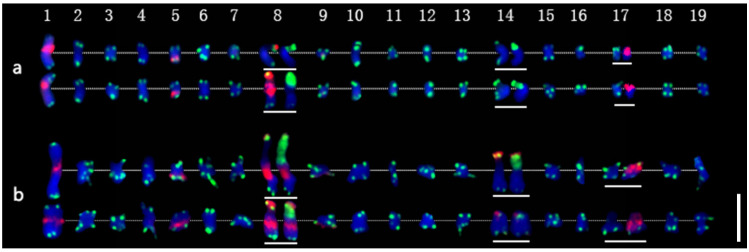
Molecular karyotypes of *P. xiaohei* and the derived doubled haploid. (**a**) *Populus simonii × P. nigra*; (**b**) Doubled haploid *Populus simonii × P. nigra*. Chromosomes 1–19 are arranged from left to right, and homologous or homeologous chromosomes are arranged from top to bottom. In the chromosomes with white lines below, the first FISH result is shown on the left (same as in Figure 3b,e), and the second FISH result is the chromosome shown on the right (same as in Figure 3c,f). Bar = 5 μm.

**Figure 5 ijms-22-11424-f005:**
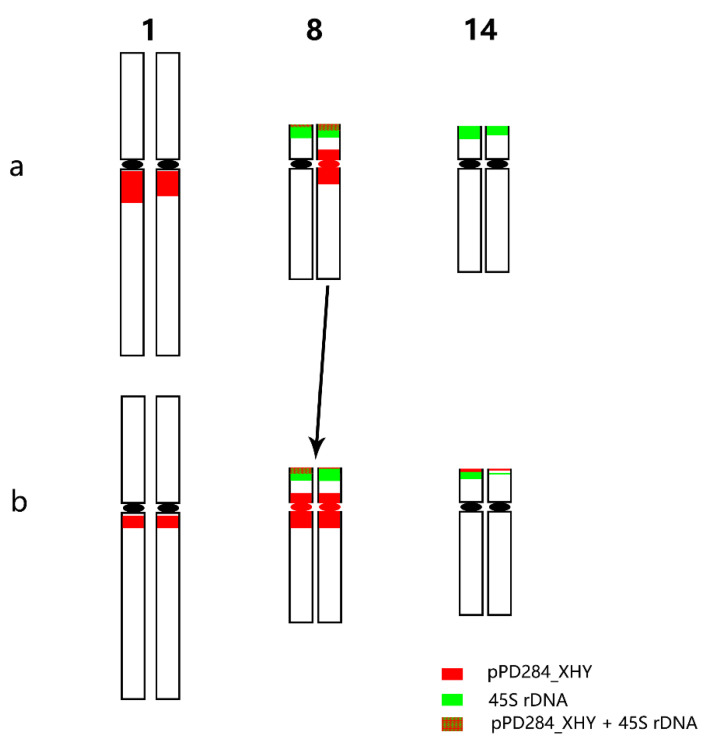
Predicted FISH pattern differences between *P. xiaohei* (**a**) and the derived doubled haploid (**b**).

**Table 1 ijms-22-11424-t001:** Information of two candidate repeats in *P. xiaohei*.

Name	Length (bp)	Sequences
pPD349_XHY	129	GTCTCTCGTAACCAGAGTCCCGGTTCGAAAGGGCCATAACTTTTGATCCGACCGTTGGATCTCCCTTAAATTTTTACAGGAGTTTCCGGACGCTGTTTTCCTTGGAGTAGATGTGGAATCGCTACTCGG
pPC284_XHY	109	CGGTTCGAAAGGGCCATAACTTTTGATCCGACCGTTGGATCTCCCTCAAATTTTCACAGGAGTTTCCGGACGCTGTTTTCCTTGGAGTAGATGTTGAATCGCTACTCGG

## Data Availability

Not applicable.

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
