# Peer review of "Molecular Karyotyping on Populus simonii × P. nigra and the Derived Doubled Haploid"

_ijms, 2021, doi:10.3390/ijms222111424_

Round 1

Reviewer 1 Report

this is an interesting and well performed study.

its most important issue is the way of the presentation - missing words, missing articles, etc.

another issue is that despite having sequenced the genome, the study reads like a rather average merely cytogenetic study, which is a pity. the 2 newly described repetitive sequences is not enough regarding that n=19 and rDNAs do not provide much information resolving the individual chromosomes. there should have been much more repeats utilized to individualize also the remaining chromosome pairs.

+correct the following:

line 48 "anther"

line 54 "basic chromosome number" to haploid chromosome number

Author Response

Dear Reviewer,

We sincerely thank you for thoroughly examining our manuscript and providing very helpful comments to guide our revision. This manuscript has been revised extensively according to the reviewer's constructive suggestions. There are three major modifications in the revised version. First, we’ve added more information about the DH plants, including more detailed explanations of the tissue culture process and the advantages of DH plants. Second, more details about the FISH process have been included in the M&M section. Third, the English language and style of the manuscript has been improved with the help of English speaking editors at AJE and “English editing” services of MDPI. The responses to the comments are given below.

Point 1: Its most important issue is the way of the presentation - missing words, missing articles, etc.

Response 1: Thank you for your comment. The English language and style of the manuscript has been carefully improved throughout the manuscript with the help of English speaking editors at AJE and “English editing” services of MDPI.

Point 2: Another issue is that despite having sequenced the genome, the study reads like a rather average merely cytogenetic study, which is a pity. The 2 newly described repetitive sequences is not enough regarding that n=19 and rDNAs do not provide much information resolving the individual chromosomes. There should have been much more repeats utilized to individualize also the remaining chromosome pairs.

Response 2: We completely agree with the reviewer on this issue. In this study, we used a documented repetitive sequence (145 bp tandem repeat) (Rajagopa et al. 1999) for homology search against the genome sequences. The 145 bp tandem repeat sequence was the first reported repeat in Populus by molecular characterization. The chromosomal distribution of 145 bp was unknown. It is conserved across different Populus species and occupies a large proportion of the genome. Since the examined P. xiaohei is a hybrid between different species, the 145 bp tandem repeat sequence might be an ideal chromosomal marker for chromosome identification, and abundant occupation of the genome may bring about the wide distribution in different chromosomes. However, 145 bp was revealed in limited chromosomes in this study. Nevertheless, the chromosomes of the P. xiaohei and the derived DH line were not wholly identified by the used FISH probes; the basic cytological features of the above plants were still relatively revealed. As you suggested, other more repetitive probes still need to be explored in the future for detailed chromosomal structure and variation analysis in P. xiaohei and the derived DH plants. Next, we will design specific probes between homologous chromosomes to better verify the homologous chromosome recombination events during meiosis of P. xiaohei. However, it is beyond the scope of the present manuscript. To help readers more clearly understand this issue and our future research, we also added some explanations at the end of the Discussion (lines 270–274).

Point 3: Line 48 "anther".

Response 3: We’ve changed "anther cultures" to "anther" (line 54).

Point 4: Line 54 "basic chromosome number" to "haploid chromosome number".

Response 4: We’ve changed " basic chromosome number " to " haploid chromosome number " (line 60).

More detailed modifications are highlighted in red in the PDF file. We sincerely hope that this revised manuscript has addressed all your comments and suggestions. We appreciate your work and hope that the corrections will be met with approval. Once again, thank you very much for your comments and suggestions.

Reviewer 2 Report

General comments:

              The authors showed the differences in the number and sensitivity of 5S rDNA, 45S rDNA and repetitive sequence signals between P. xiaohei and doubled haploid plants. The obtained results showed that the chromosomal variation within or between species can be revealed by the suitable repeat markers. In addition, the genetic mechanisim of chromosomal variation between P. xiaohei and derived doubled hapoid plants has been partially discovered.

Detailed comments and suggestions:

  1. The main aim of this study was to investigate the chromosomal variation between P. xiaohei and derived doubled hapoid plant by subsequent multicolor-FISH. However, the authors should give more information about the DH plants were used for this study (the DH plant was chosen due to its specific charateristics compared to other DH plants?)
  2. The authors should give more detail explanations on the tissue culture process (“anther – callus – embryogenesis/organogenesis – plantlet” or “anther – embryogenesis- plantlet” patways) because the somaclonal variation rate depended on the culture patway.
  3. The authors should describe more details for the reporbing process in M&M part.
  4. Data was collected from how many mitotic spreads to exclude the somaclonal variation during tissue culture that resulted in chromosomal variation?
  5. Line 44: should read “maternal” or “mother plant” (instead of “mother parent”)
  6. Line 44: should read “paternal” or “father plant” (instead of “father parent”)
  7. Line 103: Blastn should be “BLASTN”
  8. Line 110: the first FISH experiment
  9. Line 115: for the second round of FISH
  10. Line 116: on the same “mitotic spread” instead of “slide”
  11. Line 136 (Fig 3). P. xiaohei (upper panel) and the derived double haploid (lower pannel)

Conclusion:

              The manuscript could be accepted after thorough revision according to recommendations of reviewers and editors.

Author Response

Dear Reviewer,

We sincerely thank you for your thoroughly examining our manuscript and providing very helpful comments to guide our revision. This manuscript has been revised extensively according to the reviewer's constructive suggestions. There are three major modifications in the revised version. First, we’ve added more information about the DH plants, including more detailed explanations of the tissue culture process and the advantages of DH plants. Second, more details about the FISH process have been included in the M&M section. Third, the English language and style of the manuscript has been improved with the help of English speaking editors at AJE and “English editing” services of MDPI. The responses to the comments are given below.

Point 1: The main aim of this study was to investigate the chromosomal variation between P. xiaohei and derived doubled haploid plant by subsequent multicolor-FISH. However, the authors should give more information about the DH plants were used for this study (the DH plant was chosen due to its specific characteristics compared to other DH plants?)

Response 1: Haploid is the quality of a cell or organism having a single set of chromosomes, and doubled haploids (DHs) are plants derived from a single pollen grain and doubled artificially to form homozygous diploids. They all have the characteristics of homozygosity but generally weak growth. Haploid and doubled haploid plants play an important role in breeding, genetics, molecular biology, and genomics. Additionally, haploid and doubled haploid plants are adapted for mutagenesis and genetic transformation experiments, presenting the advantage of immediate production of homozygous lines. All these advantages are the reasons we decided to produce haploid or DH plants for our experiments. In fact, it is very difficult to produce haploid or DH trees. Additionally, their long-term survival is also a big problem. At the beginning of our experiment, more than 30 haploid calli lines were obtained, but the DH line used in our experiment was the first line to regenerate complete plants that grew relatively normally. Therefore, this DH plant was naturally selected for our experiment. We have included more detailed information about this question in the revised manuscript (line 47–53).

Point 2: The authors should give more detail explanations on the tissue culture process (“anther – callus – embryogenesis/organogenesis – plantlet” or “anther – embryogenesis- plantlet” pathways) because the somaclonal variation rate depended on the culture pathway.

Response 2: Thanks for your kind suggestions. The process of tissue culture went through “anther –callus – organogenesis – plantlet” pathways. In the experiment, we reduced the concentration of plant hormones and shortened the culture cycle as much as possible. More details have been provided in the “4.1 Plant materials” section (lines 279–288).

Point 3: The authors should describe more details for the reporbing process in M&M part.

Response 3: Thanks for your kind suggestion, which is valuable for improving the accuracy of the manuscript. More details have been included in the “4.5. Fluorescence in situ hybridization” in the M&M part (lines 324–339).

Point 4: Data was collected from how many mitotic spreads to exclude the somaclonal variation during tissue culture that resulted in chromosomal variation?

Response 4: We collected a total of 44 mitotic spreads of P. xiaohei and 57 mitotic spreads of DH plants. No chromosome variations were observed in different cells in the individual. As you pointed out above, we also tried our best to avoid the impact of somatic variation caused by tissue culture on the results. However, if this variation comes from before differentiation, we can only wait for more DH cell lines to produce regenerated plants for verification (lines 234-237).

Point 5: Line 44: should read "maternal" or "mother plant" (instead of "mother parent").

Response 5: We have fixed the error (line 44).

Point 6: Line 44: should read “paternal” or “father plant” (instead of “father parent”).

Response 6: We have fixed the error (line 45).

Point 7: Line 103: Blastn should be “BLASTN”.

Response 7: We have fixed the error (line 110).

Point 8: Line 110: the first FISH experiment.

Response 8: We have fixed the error (line 117).

Point 9: Line 115: for the second round of FISH.

Response 9: We have fixed the error (line 123).

Point 10: Line 116: on the same “mitotic spread” instead of “slide”.

Response 10: We have fixed the error (line 123).

Point 11: Line 136 (Fig 3). P. xiaohei (upper panel) and the derived doubled haploid (lower panel).

Response 11: Thank you for pointing out this problem in our manuscript. We have updated “P. xiaohei (upper panel) and the derived doubled haploid (lower panel)” in lines 158-159 (Fig 3).

More detailed modifications are highlighted in red in the attachment PDF file. We sincerely hope that this revised manuscript has addressed all of your comments and suggestions. We appreciate your work and hope that the corrections will be met with approval. Once again, thank you very much for your comments and suggestions.

Round 2

Reviewer 2 Report

This revised manuscript has addressed all of my comments and suggestions.

Only one small question: did you check the metaphase spread before reprobing to ensure the old signals were totally removed. In my experience, 2xSSC solution only is not enough to remove the signals. 

Author Response

Dear Reviewer,

We sincerely thank you for your thoroughly review our manuscript again. The references section of this manuscript has been revised slightly according to IJMS format requirements. The responses to the comments are given below.

Point 1: Did you check the metaphase spread before reprobing to ensure the old signals were totally removed. In my experience, 2xSSC solution only is not enough to remove the signals.

Response 1: Surly, it is crucial to avoid the signals retaining after the first round of hybridization for reprobing. To resolve it, we conducted chromosome preparation re-denature after the first round of microscopic examination (line 337). Though longer time of re-denature can remove signal more througly, it also can make chromosomes ugly. However, the process used in our study still may removed the old signals largely. Each preparation was checked to make sure that the old signals were removed maximally, and the slides with the least old signal for the next reprobing progress. Futhermore, to make sure the signals accurately identified from the two rounds hybridization, we conducted the only one round of hybridization using the probes combination, though we did not mention it in the manuscript. It assisted to determine the chromosomal distributions of the used FISH probes.

More revised details can be viewed using the “Track Changes” function of MS Word in the file below (antuor-coverletter-15142676.v1.docx). We appreciate your work and thank you very much for your comments and suggestions.
